

# Fuzzy multi-objective optimization model to design a sustainable closed-loop manufacturing system

Sajida Kousar[1], Asma Alvi[1], Nasreen Kausar[2], Harish Garg[3], Seifedine Kadry[4,5] and Jungeun Kim[6]

[1] Department of Mathematics and Statistics, International Islamic University, Islamabad, Pakistan
[2] Department of Mathematics, Faculty of Arts and Sciences, Yildiz Technical University, Istanbul, Turkiye
[3] Department of Mathematics, Thapar Institute of Engineering & Technology (Deemed University), Patiala, Punjab, India
[4] Department of Computer Science and Mathematics, Lebanese American University, Beirut, Lebanon
[5] MEU Research Unit, Middle East University, Amman, Jordan
[6] Department of Computer Engineering, Inha University, Incheon, Republic of South Korea

Corresponding authors
Sajida Kousar,
sajida.kousar@iiu.edu.pk
Jungeun Kim, jekim@inha.ac.kr

## ABSTRACT

Republicans and Democrats practically everywhere have been demonstrating concerns about environmental conservation to achieve sustainable development goals (SDGs) since the turn of the century. To promote fuel (energy) savings and a reduction in the amount of carbon dioxide $CO_2$ emissions in several enterprises, actions have been taken based on the concepts described. This study proposes an environmentally friendly manufacturing system designed to minimize environmental impacts. Specifically, it aims to develop a sustainable manufacturing process that accounts for energy consumption and $CO_2$ emissions from direct and indirect energy sources. A multi-objective mathematical model has been formulated, incorporating financial and environmental constraints, to minimize overall costs, energy consumption, and $CO_2$ emissions within the manufacturing framework. The input model parameters for real-world situations are generally unpredictable, so a fuzzy multi-objective model will be developed as a way to handle it. The validity of the proposed ecological industrial design will be tested using a scenario-based approach. Results demonstrate the high reliability, applicability, and effectiveness of the proposed network when analyzed using the developed techniques.

## INTRODUCTION

It is an eminent fact that major environmental concerns have arisen because of global megatrends such as the exponential growth of population and pollution due to the disposal of untreated and unavoidable manufacturing waste, carbon dioxide $CO_2$ emissions, energy consumption, and depletion of natural resources. Manufacturing companies represent the backbone of contemporary industrialized civilization and have long been regarded as the fulcrum of the global economy. It has been indicated by *Karuppiah, Sankaranarayanan & Lo (2024)* that advancement in industrialization is adversely impacting the environment and human life on earth but it is yet uncertain how to evaluate their long-term viability.

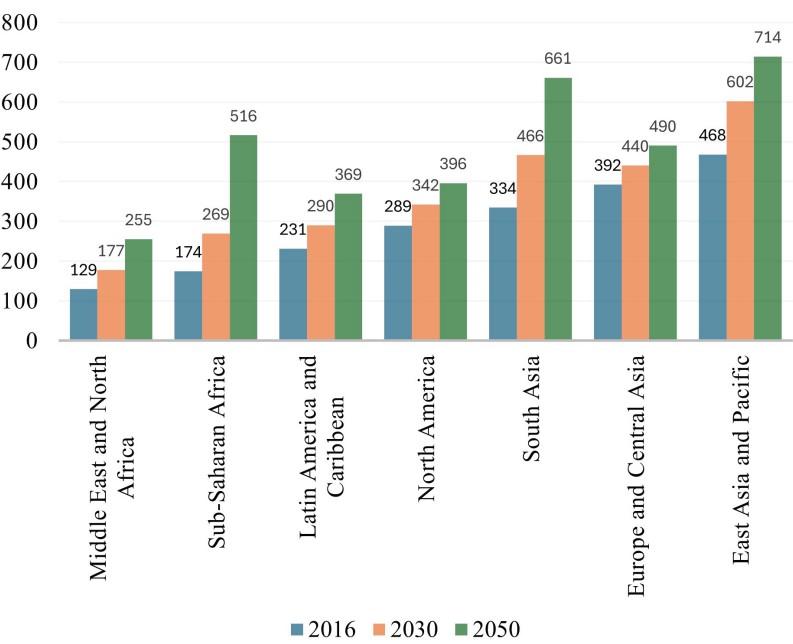

**Figure 1 Projected waste generation (million tons per year) by region.**

*Sheng et al. (2023)* identified that the manufacturing sector is a major contributor to huge energy consumption, waste generation, and greenhouse emissions, therefore researchers are required to propose/develop such kind of industrial model that may ensure enhanced resource efficiency, minimize waste, and reduce carbon emission. The waste generation and carbon emission rate in Fig. 1 (*Kaza et al. (2018)*) and Fig. 2 (*IEA (2023)*) indicate the scale of environmental degradation caused by these factors. Although new terms like environmentally conscious manufacturing, green manufacturing, and reverse manufacturing are thought to be connected to sustainability, there is a wide range of views and interpretations, due to which it has become difficult to find a definitive definition for sustainability in manufacturing firms (*Sartal et al., 2020*).

It is incumbent that industrial communities develop such modern enterprises which may ensure that enacting a sustainable system is an essential requirement (*Anwar et al., 2023*). Most definitions of sustainability include terms like green, 'clean, keep, retain, stability, ecological balance, natural resources, and the environment. It can be difficult to define the term "sustainability," usually it is understood as a way of nonlethal manufacturing products, and natural resources preservation, and frugal. In the meantime, there is still a disagreement over what constitutes and defines sustainability; promoting sustainable development has emerged as a significant worldwide goal. The phrase sustainable development (SD) is widely used to describe contemporary ideas of sustainability. SD has been regarded as a form of development that can gratify the needs of human beings devoid of jeopardizing the capability of future generations to gratify their desires. It is crucial to consider SD principles in every activity or choice made to achieve responsible and sustainable development. One of the most crucial topics for advancing the

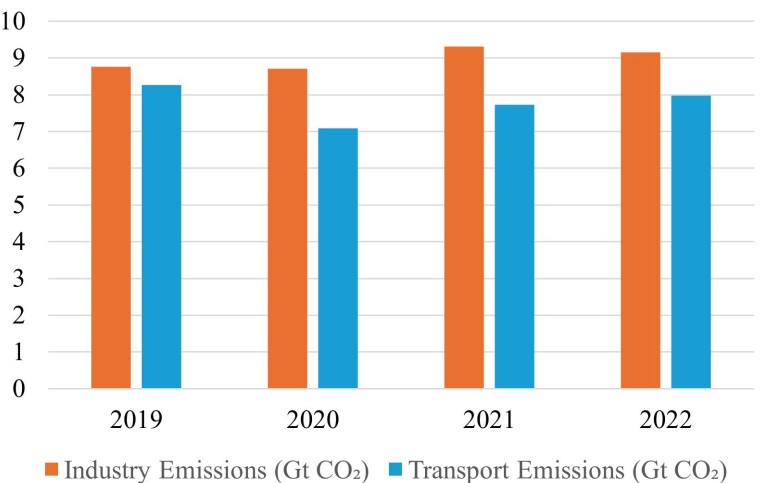

**Figure 2 Carbon emission from industrial and transportation sector.**

**Table 1 Seventeen sustainable development goals (*UN, 2015*).**

| | | | |
|---|---|---|---|
| 1. No poverty | 2. Zero hunger | 3. Good health and well-being | 4. Quality education |
| 5. Gender equality | 6. Clean water and sanitation | 7. Affordable and clean energy | 8. Decent work and economic growth |
| 9. Industry, innovation and infrastructure | 10. Reduced inequalities | 11. Sustainable cities and communities | 12. Responsible consumption and production |
| 13. Climate action | 14. Life below water | 15. Life on land | 16. Peace, justice and strong institutions |
| 17. Partnership for the goals | | | |

overall goal of SD is sustainable manufacturing. Designing for sustainability and sustainable development becomes the design of sustainable manufacturing firms. To accomplish sustainability, it is significant to appreciate the models and interrelated needs sustainably.

Over the last century, sustainability has been attracting both researchers and the industrial sector from almost all over the world (*Bello-Pintado, Machuca & Danese, 2023*). To start along a more sustainable road, several objectives and targets have been established. To encourage the concept of sustainability by confronting the three primary global concerns, the United Nations has defined seventeen sustainable development targets (SDGs) (see Table 1). The goals for sustainable development are interconnected, which means that achieving one requires addressing problems that are frequently linked with another. Just as the unadorned stressful situation on earth is caused by numerous interconnected global megatrends, so are the goals for sustainable development.

One of the important goals of SDGs is SDG9 which focuses on industry, innovation, and infrastructure. Here, it is important to mention that an important element of SD is sustainable industrialization. In developing countries, industrialization has produced several environmental impacts and sustainable industrialization is being promoted at a

rapid pace; as it has been indicated by *Chen et al. (2024)* that due to it there is an increase in carbon emission in many countries. Regarding this, action has been taken on the defined principles to encourage fuel (energy) saving and the reduction in the emissions of $CO_2$ in many manufacturing industries. The expanded efforts (cost, energy, and emissions) of $CO_2$ at all levels and elements, from the process, product, and the linked system through the complete product life cycle, must be made to make a manufacturing firm more sustainable. It is necessary to define a sustainable manufacturing system to minimize environmental effects. Utilizing mathematical optimization approaches, the issues with sustainable industrialization are resolved. When solving problems with sustainable manufacturing, many goal functions must be taken into account at once. For instance, in the sugar sector, boosting sugar output while decreasing waste is a problem. Because of this, no one solution can be referred to as the best and a variety of trade-off options must be taken into account. These issues serve as illustrations of the class of optimization problems known as multi-objective optimization problems.

To handle the range of optimization approaches used in real-world circumstances, multi-objective optimization is crucial. For addressing several objectives and offering solutions that are more in line with system requirements, there are numerous specialized strategies. The construction of the non-inferior (Pareto optimum) solution is a general objective in multi-objective optimization due to the multi-criteria of such issues; a non-inferior solution is one in which an improvement in one objective necessitates the deterioration of another. Typically, several trade-off solutions (referred to as the Pareto set) are discovered using the stated concept of optimality. Some data used for analysis and other purposes in the actual world are questionable.

However, decision-makers must take this unpredictability into account while designing their networks. The creation of a multi-objective optimization helps to address uncertainty resulting from the impreciseness of data. Fuzzy multi-objective optimization is a method that relies only on the formulation of fuzzy information in terms of membership functions to address the optimality of the fuzziness model utilizing currently available multi-optimization tools and methodologies (*Paksoy, Pehlivan & Özceylan, 2012*).

Supply chain operations now inevitably influence several company sectors and international trade. Customers play a fundamental part in these processes, and markets work to meet their needs by taking lead time minimization, product quality improvement, quick delivery, and their prospective expectations into account. The lack of precious resources, the state of the economy, social and environmental effects, and the rise in customer requests aimed at the finest products, lead policymakers towards deliberation and reverse logistics networks (*Soon et al., 2022*; *Kazancoglu et al., 2022*). Reverse logistics advantages in this area let organizations best the most of their businesses through material recycling, return, and reuse in multi-level systems that help improve the environment (*Tavana et al., 2022*).

The modern closed-loop manufacturing system will unavoidably have an impact on several industrial sectors as well as global trade. The primary goal of the research is to minimize the cost, total energy consumption, and overall $CO_2$ emissions of a suggested closed-loop manufacturing system (CLMS) by choosing the overall quantity of existing

services and significantly reducing the overall cost within multiple levels. Each place in this design seems to have a limited scope, and other places for each place are defined. The supply rates to every type of customer are stated for each period. Each warehouse has a certain capacity, and after each period, the inventory cost is taken into consideration for the items that are still supplied. Since the multi-objective issue has been taken into account, the goal of the research is to use optimization approaches to validate the model and produce the best outcomes. The closed-loop supply chain (CLSC) is thought to be a tactic for developing a sustainable supply network, as *Roudbari, Fatemi Ghomi & Eicker (2024)* stated that it can play two major roles; satisfying customers' demands and effectively recapturing the value of the returned product. The following are the primary goals of the research in the manufacturing system:

1) Constructing a closed-loop, sustainable supply network for the manufacturing phase in the industrial sector.
2) Adopting a linear programming methodology to reduce the overall expenses of the manufacturing logistics chain.
3) To supply materials and transportation of returned goods for subsequent use, take into consideration forward and reverse logistics.

The methodology initially evaluates the various providers under sustainable standards. The network of closed-loop supply chains is assessed to get the best possible amount from the potential providers chosen based on sustainable standards to satisfy consumer requests. Supplier (SU) then ships the quantity $q_{SU.PL}^{Raw}$ through plant (PL) to retailer (RE), and then transport it to customer (CU). The product as waste material is collected by collection center (CC) by CU and transported to disassembly center (DC) to refurbishing center (RC) back on the road to PL using various modes of transportation. The next section of this study presents a comprehensive literature review along with the current study's motivation, aim, and objectives. "Closed-Loop Manufacturing System Model Formulation" develops the thorough aspects of dilemma formulation, taking into consideration objective function and problem constraint. "Solution Methodology" presents the solution methodology followed by a case study, results, comparison of techniques, and conclusion.

## LITERATURE REVIEW

There have been several studies in supply chain systems, closed-loop supply chains, logistics system planning, and remanufacturing (*Lo et al., 2021*; *Nunes, Causer & Ciolkosz, 2020*; *De & Giri, 2020*). However, there are relatively few chronological writings on manufacturing supply chains. Initially, *Lim et al. (2021)* offers industrial sector approaches and created supply chain topics to draw scholars by providing an example of a specific supply chain network. *Marić & Opazo-Basáez (2019)* supports sustainable industrialization and ensures that providing an industrial supply chain for recycling products, results in more quality in the manufacturing industry. The planned measure was then given environmental and economic investigation in the same study. A detailed

analysis of industrial practices and approaches for a design intended for sustainable actions is presented by *Shekarian et al. (2022)*. They provided a new and enhanced classification of the work indicating more routes so it may be used to analyze how different firms reach sustainability.

The implementation of systems for closed-loop supply chain processes and reverse logistic (RL) has been studied in depth by a significantly as well as an increasing number of studies. RL is a mechanism in which products and other products are moved from their ultimate place to further destinations to be recycled, disposed of, or used in other operations (*Fathollahi Fard & Hajaghaei-Keshteli, 2018*; *Cheraghalipour, Paydar & Hajiaghaei-Keshteli, 2017*). Since RL promotes better growth and worth including both company owners and consumers, it has been adopted by the majority of emerging nations in various industries and enterprises (*Hsu, Tan & Mohamad Zailani, 2016*).

Keeping in view these advantages, previous research utilized reverse or closed-loop supply chain systems in their networks (*Ritchie et al., 2000*; *Govindan, Soleimani & Kannan, 2015*), and discussed closed-loop supply chain systems reversal logistics, too. As in research, they researched different aspects which include ecological factors, economic and legal aspects, and their positives from a supply chain and CLSC perspective, *Özceylan, Paksoy & Bektaş (2014)* suggesting a nonlinear model for constructing a closed-loop supply system in the accurate economic climate for balance separating and supply lines. The levels in the forward process include supplier, plant, retailer, and customer also the same number of levels are in the design of the reverse process which are the refurbishing center, waste center, disassembly center, and collecting center. To fulfill the need for product returns, certain writers, including *Keyvanshokooh, Ryan & Kabir (2016)*, examined and improved the CLSC and presented models. The model used Benders' approach with hybrid probabilistic and resilient programming to get the best solution to the issue. An increasing tendency towards networks of sustainability practices may be seen throughout the development of supply chain management. *Rahemi et al. (2020)*, *Li, Manier & Manier (2019)*, *Balcombe et al. (2018)* and *Soleimani et al. (2017)* included sustainability, optimization, and new distinct networks. They used multi-objective programming to construct a trustworthy and fuzzy CLSC that takes into account uncertainty. Suppliers, manufacturers, distributors, warehouses, return centers, and disposal facilities are all part of this supply chain. The maximization of overall revenue, taking sustainability principles, and enhancing customer response are examples of objective functions. To enhance the outcomes, a genetic algorithm (GA) was utilized. *Devika, Jafarian & Nourbakhsh (2014)* suggesting a linear model creates a closed-loop supply chain that is sustainable; to concurrently research the impacts of the environment, society, and economy in a specific context. The design contains six stages of collecting, recycled, reconstruction, refinement, waste, and supplementary consumers in the reverse direction, and four stages of supplies, manufacturing, distributing, and consumers in a forward manner. To cut carbon emissions, *Talaei et al. (2016)* created a closed-loop, environmentally friendly model of the supply chain. The suggested model tackles the position, in addition to optimal allocation problems, and has two objectives. The authors combined fuzzy sets and robust programming to deal with the uncertainty.

To solve the model, the epsilon constraint approach is employed. By creating a CLSC network and taking into account both optimizing profitability and social capacities by recycling items *Pishvaee & Torabi (2010)* and *Panda, Modak & Cárdenas-Barrón (2017)* attempted to maximize their suggested model by offering a bi-objective design framework about a closed-loop supply chain in an ambiguous setting. Although the secondary purpose seeks in minimizing the time needed to receive goods and services, the first goal seeks to decrease expenses. The model contains four layers of collecting, rebuilding, recycled, and third consumers in both forward and reverse directions, three phases of manufacturing, distribution, and consumers are also included. *Soleimani & Kannan (2015)* constructed a CLSC system on large systems. The suggested system is comprehensive, multi-level, multi-period, and multi-product. The model is solved using the GA, particle swarm optimization (PSO) technique, hybrids genetic algorithm particle swarm optimization (GAPSO), and the exact approach (CPLEX). *Kaya & Urek (2016)* created a CLSC network to place stocks and set prices in the best possible places. To find services and inventory, a nonlinear mixed integer programming approach is suggested in this study. The goal of the problem was to increase the suggested supply chain network's overall profit.

While exploring the design of reversible logistics networks, taking into account multiple layers including warehouse, transit, logistic, manufacturer, distributor, and client, it was seen how each stage interacted with the supply chain (*Fulconis & Philipp, 2018*). *Sadeghi, Mina & Bahrami (2020)* used mixed integer linear programming (MILP) to create a reverse chain for industrial logistics. The levels of production, distribution, separation, and destruction were among those that were taken into account. To further clarify the problem, they developed a multi-period, multi-product model. Reverse supply networks' environmental properties and their supply chain were analyzed and categorized by *Islam et al. (2021)*. *Govindan & Soleimani (2017)* presented a review of reverse logistics and other earlier research in this field. *Kazemi, Modak & Govindan (2019)* developed evaluations of the work on reverse logistics and CLSC as well as a detailed analysis of content. For armed hardware including replacement parts, a closed-loop system was developed (*Wang & Shi, 2019*). Since it is essential to get to the military supplies, the target function of the minimal durations is used in this research's design. To handle the suggested system, the Lion meta-heuristic method is employed. *Torabi et al. (2016)* created a reverse supply chain while taking the disrupting risk and operating limits into account. The suggested model is then solved using based on fuzzy logic programming, and even the goal function of minimizing cost is then optimized. *Gholizadeh et al. (2022)* added the creation of a renewable system to the list of priorities while utilizing reverse logistics to address both environmental and economic concerns. In this study, reproduction and recycling levels are used to reduce the manufacturing cost's objectives functions. *Zhang et al. (2021)* proposes a problem with logistics management and choice in the production chain with an emphasis on the criteria for choosing the best RL providers. The best RL logistics provider for remanufacturing is suggested in this scheme, and the strategy is resolved using a multi-criteria decision-making (MCDM) approach that includes an analytical hierarchy process (AHP) also a technique for order of preference by similarity to the ideal solution (TOPSIS). *Pourjavad & Mayorga (2019)*, *Pourmehdi, Paydar & Asadi-Gangraj (2020)*, *Gholizadeh & Fazlollahtabar (2020)*,

**Table 2 Literature reivew.**

| Study | Model | Objective functions | Solution approach |
|---|---|---|---|
| *Akbari-Kasgari et al. (2022)* | Mixed integer linear programming | Maximize profit, minimize water consumption and air pollutants, and maximize social desirability | ε—constraint method, weighted sum method |
| *Goodarzian et al. (2023)* | Mixed integer linear programming | Minimize cost, minimize environmental impacts and Optimize social factors | ε—constraint method |
| *Lotfi et al. (2024)* | A two-stage robust stochastic multiobjective programming mode | Minimize cost, $CO_2$ emissions, energy consumption and maximising employment | Lp-metric method |
| *Khalili-Fard et al. (2024)* | A novel two-stage stochastic model | Minimize cost, maximize the remaining fuel and maximize total job opportunities | Multi-objective particle swarm optimization |
| *Gholipour et al. (2024)* | Multi-objective optimization method | Minimize cost, minimize risk and maximize accountability | A meta-heuristic algorithm |
| *Momeni, Jain & Bagheri (2024)* | Multi-objective mixed-integer programming | Minimize cost, minimize environmental impact and social impact | ε—constraint method |
| Present study | Multi-objective mixed-integer programming | Minimize total investment cost, minimize energy consumption, and minimize carbon emission | Four-valued refined neutrosophic optimization technique |

*Pahlevan, Hosseini & Goli (2021)*, *Akbari-Kasgari et al. (2022)*, *Amirian, Khalili & Mehrabian (2022)* are some investigations on CLSC or RL in industrial manufacturing systems. Even if some earlier research created industrial manufacturing systems, there have been relatively few investigations on CLSC or RL for such items. Table 2 contains some studies related to sustainable industrialization.

The previous studies on CLSC and RL have mostly focused on developing new systems for selling rebuilt goods, recycling product returns, and converting waste into renewable energy. Although these efforts have expanded global attention, more comprehensive approaches toward closed and reverse supply chains, beyond just economic objectives, are necessary. In addition to addressing climate change, developing a holistic CLSC system presents wide-ranging opportunities for investors to maximize the value of products across their entire life cycle. Over the past two decades, significant developments in CLSC systems have simplified policymaking for addressing complex issues such as re-manufacturing used goods, meeting customer demand, and identifying marketing strategies for re-manufactured products. However, much of the existing research has focused on the forward supply chain and its role in meeting client demands, resulting in higher revenues and lower expenses for suggested systems. While such insights are valuable, they overlook critical aspects of closed-loop and backward logistics systems that are key to achieving a circular economy.

Despite the growing research on supply chain management, there is a noticeable gap in research addressing the full spectrum of CLSC systems. Specifically, the integration of circular economy principles, the development of multi-objective optimization models that consider both economic and environmental sustainability, and the need for stakeholder collaboration remain under-explored. Additionally, while most research has concentrated on the manufacturing industry, other sectors such as electronics, textiles, and consumer goods lack detailed studies on implementing efficient backward logistics systems.

Additionally, the advent of digital transformation, AI, and IoT technologies presents new challenges and opportunities for enhancing the efficiency of CLSC networks. However, the role of these technologies in building resilient CLSC systems remains insufficiently explored in the current literature. Addressing these gaps could provide a more comprehensive framework for the implementation of CLSC, promoting sustainability and long-term profitability in various industries. Therefore, the following represent the key goals and suggestions to address the study gaps throughout the research:

1) Literature indicated that there are plenty of research studies on aforementioned topic in industrial supply chains (ISC), therefore, this study begins by examining the existing research on supply chain system architecture for the industrial sector.

2) Here on basis of its creative system, the present investigation has created revolutionary modeling. Although the majority of earlier studies recommended forward supply chain systems.

3) The suggested framework not only includes a tire backward route to preparing the gathered produced goods again for the secondary market, but also minimizes the total costs, overall energy, and $CO_2$ emissions in both forward and backward reverse directions.

4) Because of its distinctive system, the research presents a new mixed integer linear programming (MILP) approach from a mathematical standpoint.

5) The objective will be achieved using a four-valued refined neutrosophic optimization approach.

## CLOSED-LOOP MANUFACTURING SYSTEM MODEL FORMULATION

The imperative for returned products from the manufacturing process is one of the emerging concerns in CLMS design for the industrial system. These materials can not only be obtained as raw materials in a range of industries and enterprises but they can also be recycled and used again. In this study, a CLMS (see Fig. 3) is developed to minimize total costs, energy use, and $CO_2$ emissions in the manufacturing system's supply chain while taking into account returned goods on a reverse flow. Figure 3 encapsulates a schematic design of a CLMS. It moves ahead with production from the facility directed through the retailer to customers before going backward with the collection, disassembly, and refurbishment centers. During these phases, the recycling process is carried out to change the commodities into new ones. Our suggested model incorporates tactical decisions, such as production, inventory planning, capacity management, and transportation system, as well as strategic considerations, such as the best choice of facility sites made up of disassembly centers, recycling facilities, and collection centers. The locations of the factory and customers are taken for granted.

The following are the fundamental presumptions of the model:

1) The prospective sites of each center (Plant, Retailer, Collection center, Disassembly center, waste center, and Refurbishing center) are known and each has a limited capacity.

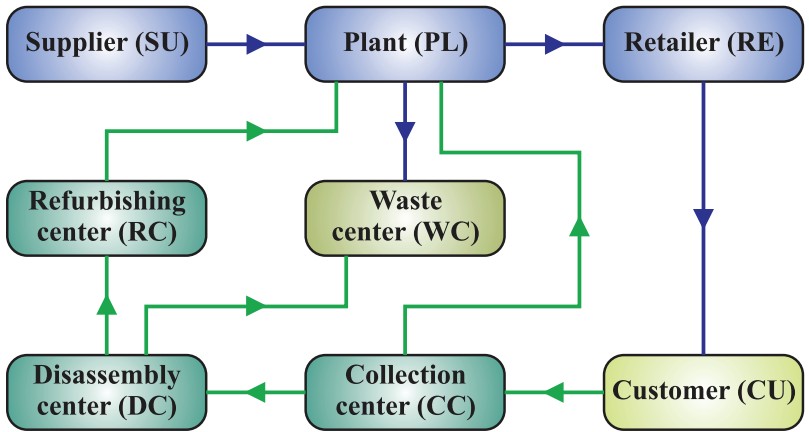

**Figure 3** Structure of the proposed closed-loop manufacturing system.

2) The predicted customer demand rate for each time.

3) There cannot be a shortage.

4) The inventory cost for the remaining products is calculated at the end of each period.

5) All costs are taken into account after each session.

## Decision variables and parameters

**Notations**

| | |
|---|---|
| **SU** | supplier |
| **PL** | plant |
| **RE** | retailer |
| **CU** | customer |
| **CC** | collection center |
| **DC** | disassembly center |
| **WC** | waste center |
| **RC** | refurbishing center |
| $k_{PL_j}$ | number of processes involved in the plant; $j \in \{1, 2, \ldots, k_{PL}\}$ |

**Parameters**

| | |
|---|---|
| $C_{CL}^{Fixed}$ | Fixed cost (\$) of closed loop system |
| $C_{SU.PL}^{Raw}$ | Raw material cost (\$) |
| $C_{SU}^{Raw}$ | Unit raw material cost (\$) in SU |
| $C_{PL.RE}^{Product}$ | Manufactured product cost (\$) |
| $C_{PL}^{Product}$ | Unit manufactured product cost (\$) |
| $C_{PL.RE}^{I}$ | Inventory cost (\$) from a PL to a RE |
| $C_{RE}^{I}$ | Unit inventory cost (\$)in RE |
| $C_{CL}^{T}$ | Transportation cost (\$) of closed loop system |
| $C_{A}^{T.B}$ | Unit transportation cost (\$) (per mile) of material from $A$ to $B$ |
| $D_{A.B}$ | Distance in miles from $A$ to $B$ |

| | |
|---|---|
| $G_A$ | Quantity (in kg) of the material produced from $A$ (per month) |
| $\Re_{PL_j}$ | Machine manufacturing rate (%) involved in process $j$ |
| $\mu_{PL_j}$ | Machine efficiency (%) involved in process $j$ |
| $\rho_{PL_j}^{air\ compressor}$ | Compressed air's capacity ($m^3/h$) of compressor in process $j$ |
| $N_{A_j}^{machine}$ | Installed power (Kw) of machine in $A$ involved in process $j$ |
| $N_{A_j}^{compressor}$ | Installed power (Kw) of compressor in $A$ involved in process $j$ |
| $N_{A_j}^{airconditioner}$ | Installed power (Kw) of air conditioner in $A$ involved in process $j$ |
| $N_{A_j}^{illuminations}$ | Installed power (Kw) of illuminations in $A$ involved in process $j$ |
| $em_{PL}$ | Quantity of $CO_2$ (in kg) released from PL |
| $em^T$ | Quantity of $CO_2$ (in kg) released from vehicles during the transportation |
| $em_{A.B}^{T.Raw}$ | Quantity of $CO_2$ (in kg) emitted during transportation of raw materials from $A$ to $B$ |
| $em_{A.B}^{T.Product}$ | Quantity of $CO_2$ (in kg) emitted during transportation of manufactured product from $A$ to $B$ |
| $em_{PL_j}^{machine}$ | Quantity of $CO_2$ (in kg) released from machines involved in process $j$ |
| $em_{PL_j}^{aircompressor}$ | Quantity of $CO_2$ (in kg) released from air compressor involved in process $j$ |
| $em_{PL_j}^{airconditioner}$ | Quantity of $CO_2$ (in kg) released from air conditioner involved in process $j$ |
| $em_{PL_j}^{illuminations}$ | Quantity of $CO_2$ (in kg) released from illuminations involved in process $j$ |
| $\alpha_{PL_j}$ | $CO_2$ emission ($kg/kWh$) depending upon the energy resources that are used in a PL |
| $\alpha^T$ | $CO_2$ emission ($kg/mile$) depending upon the transportation of material |
| $\beta_{PL_j}^{machine}$ | The total waste ratio (in %) for machines involved in process $j$ |
| $\gamma_{PL_j}^{air\ conditioner}$ | Covering rate (%) for each air conditioner used in process $j$ |
| $\gamma_{PL_j}^{air\ compressor}$ | Covering rate (%) for each air compressor used in process $j$ |
| $\gamma_{PL_j}^{illuminations}$ | Covering rate (%) for each illumination used in process $j$ |
| $\gamma_A^{air\ conditioner}$ | Covering rate for each air conditioner used at $A$ (other than PL) |
| $\gamma_A^{illuminations}$ | Covering rate (%) for each illumination used at $A$ (other than PL) |
| $D_A$ | Demand (per month) of material (in Kg) at $A$ |
| $Ca_A$ | Capacity (in Kg) of the $A$ |
| $\mathbf{V}$ | Capacity in kg per vehicle |
| $R_1^{Rate}$ | Return rate (%) from customer to CC |
| $R_2^{Rate}$ | Return rate (%) from CC to DC |
| $R_3^{Rate}$ | Return rate (%) from CC to PL |
| $R_4^{Rate}$ | Return rate (%) from DC to RC |
| $R_5^{Rate}$ | Return rate (%) from DC to WC |

**Decision variables**

| | |
|---|---|
| $q_{A.B}^{Product}$ | Quantity of produced material (kg) transported from $A$ to $B$ |
| $q_{A.B}^{Raw}$ | Quantity of raw material (kg) transported from $A$ to $B$ |
| $q_{PL_j}^{Raw}$ | Quantity of raw material (kg) used in process $j$ in PL |

$q_{PL_{j+1}}^{Raw}$      Quantity of material (kg) shipped from process $j$ in PL

$n_{PL_j}^{machine}$      Number of machinery (unit) in the plant involved in process $j$

$n_A^{air\ conditioner}$      Number of air conditioners (unit) installed at $A$

$n_A^{illuminations}$      Number of illumination (unit) installed at $A$

Note: In the description, instead of writing the names of each center, the change in one center to another has been written as "from $A$ to $B$".

## Objective functions

### Total cost invested $f_1$

The total investment cost in a circular system is considered by combining the all fixed costs of a closed loop system which include costs of land, building, equipment, services, and wages; cost of raw material; cost of transportation of raw materials; costs of manufacturing and inventory. Here the total cost $f_1$ to minimize is given in Eq. (1).

$$\text{Minimize} f_1 = C_{CL}^{Fixed} + C_{SU.PL}^{Raw} + C_{PL.RE}^{Product} + C_{PL.RE}^{I} + C_{CL}^{T} \tag{1}$$

where, the fixed cost $C_{CL}^{Fixed}$ of the circular supply chain model is obtained by the combination of costs of the plant $C_{CL}^{PL}$, retailer $C_{CL}^{RE}$, collection center $C_{CL}^{CC}$, disassembly center $C_{CL}^{DC}$, waste center $C_{CL}^{WC}$, and refurbishing center $C_{CL}^{RC}$. These costs are given in Eqs. (2)–(6).

$$C_{CL}^{PL} = C_{PL}^{Land} + C_{PL}^{Building} + C_{PL}^{Equipment} + C_{PL}^{services} + C_{PL}^{Wages} \tag{2}$$

$$C_{CL}^{RE} = C_{RE}^{Land} + C_{RE}^{Building} + C_{RE}^{Equipment} + C_{R}^{services} + C_{RE}^{Wages} \tag{3}$$

$$C_{CL}^{CC} = C_{CC}^{Land} + C_{CC}^{Building} + C_{CC}^{Equipment} + C_{CC}^{services} + C_{CC}^{Wages} \tag{4}$$

$$C_{CL}^{DC} = C_{DC}^{Land} + C_{DC}^{Building} + C_{DC}^{Equipment} + C_{DC}^{services} + C_{DC}^{Wages} \tag{5}$$

$$C_{CL}^{WC} = C_{WC}^{Land} + C_{WC}^{Building} + C_{WC}^{Equipment} + C_{WC}^{services} + C_{WC}^{Wages} \tag{6}$$

The cost of one unit of raw material $C_{SU.PL}^{Raw}$ from supplier to plant and the cost of one unit of product from plant to the retailer is defined in Eqs. (7) and (8).

$$C_{SU.PL}^{Raw} = C_{SU}^{Raw} q_{SU.PL}^{Raw} \tag{7}$$

$$C_{PL.RE}^{Product} = C_{PL}^{Product} q_{PL.RE}^{Product} \tag{8}$$

Inventory cost $C_{CL.RE}^{I}$ at the retailer is calculated by using Eq. (9).

$$C_{PL.RE}^{I} = C_{RE}^{I} q_{PL.RE}^{Product} \tag{9}$$

The transportation cost of material $C_{CL}^{T}$ in the circular supply chain model is obtained by the combination of the transportation costs of material transported between various segments of CLSC as in Eq. (10).

$$C_{CL}^{T} = C_{CL}^{T.PL} + C_{CL}^{T.RE} + C_{CL}^{T.CU} + C_{CL}^{T.CC} + C_{CL}^{T.DC} + C_{CL}^{T.WC} + C_{CL}^{T.RC} \tag{10}$$

The transportation costs involved in Eq. (10) are derived from Eqs. (11)–(17).

$$C_{CL}^{T.PL} = C_S^{T.PL} \frac{q_{SU.PL}^{Raw}}{V} D_{SU.PL} + C_{CC}^{T.PL} \frac{q_{CC.PL}^{Raw}}{V} D_{CC.PL} + C_{CL}^{T.PL} = C_{RC}^{T.PL} \frac{q_{RC.PL}^{Raw}}{V} D_{RC.PL} \tag{11}$$

$$C_{CL}^{T.RE} = C_{Plant}^{T.RE} \frac{q_{PL.RE}^{Product}}{V} D_{PL.RE} \tag{12}$$

$$C_{CL}^{T.CU} = C_{RE}^{T.CU} \frac{q_{RE.CU}^{Product}}{V} D_{RE.CU} \tag{13}$$

$$C_{CL}^{T.CC} = C_{CU}^{T.CC} \frac{q_{CU.CC}^{Raw}}{V} D_{CU.CC} \tag{14}$$

$$C_{CL}^{T.DC} = C_{CC}^{T.DC} \frac{q_{CC.DC}^{Raw}}{V} D_{CC.DC} \tag{15}$$

$$C_{CL}^{T.WC} = C_{Plant}^{T.WC} \frac{q_{PL.WC}^{Product.WC}}{V} D_{PL.WC} + C_{DC}^{T.WC} \frac{q_{DC.WC}^{Raw}}{V} D_{DC.WC} \tag{16}$$

$$C_{CL}^{T.RC} = C_{DC}^{T.RC} \frac{q_{DC.RC}^{Raw}}{V} D_{DC.RC} \tag{17}$$

**Total energy consumption $f_2$**

The second objective is to minimize the total energy consumption during the production process. The objective is mathematically expressed in Eq. (18).

$$\text{Minimize} f_2 = E^{PL} + E^{RE} + E^{CC} + E^{DC} + E^{WC} + E^{RC} \tag{18}$$

In the above equation, the total energy consumed by the plant $E^{PL}$ is computed in Eq. (19).

$$\begin{aligned}
E^{PL} &= \sum_{j=1}^{k_{PL}} E_{PL_j}^{machine} + \sum_{j=1}^{k_{PL}} E_{PL_j}^{air\,copmressor} + \sum_{j=1}^{k_{PL}} E_{PL_j}^{air\,conditioner} + \sum_{j=1}^{k_{PL}} E_{PL_j}^{illuminations} \\
&= \sum_{j=1}^{k_{PL}} \left( \frac{q_{PL_j}^{Raw}}{\Re_{PL_j} \mu_{PL_j}} N_{PL_j}^{machine} n_{PL_j}^{machine} \right) + \sum_{j=1}^{k_{PL}} \left( \frac{q_{PL_j}^{Raw}}{\Re_{PL_j} \mu_{PL_j}} \frac{N_{PL_j}^{air\,compressor}}{\rho_{PL_j}^{air\,compressor}} v_{PL_j}^{air\,compressor} n_{PL_j}^{machine} \right) + \\
&\quad \sum_{j=1}^{k_{PL}} \left( N_{PL_j}^{air\,conditioner} n_{PL_j}^{air\,conditioner} \frac{q_{PL_{j+1}}^{Raw}}{G_{PL}} \right) + \sum_{j=1}^{k_{PL}} \left( N_{PL_j}^{illuminations} n_{PL_j}^{illuminations} \frac{q_{PL_{j+1}}^{Raw}}{G_{PL}} \right)
\end{aligned} \tag{19}$$

Equation (20) provides the energy consumed at the retailer's end.

$$E^{RE} = N_{RE}^{air\,conditioner} n_{RE}^{air\,conditioner} \frac{q_{PL.RE}^{product}}{G_{RE}} + N_{RE}^{illuminations} n_{RE}^{illuminations} \frac{q_{PL.RE}^{Product}}{G_{RE}} \tag{20}$$

Equation (21) provides the energy consumed at the collection center.

$$E^{CC} = N_{CC}^{air\,conditioner} n_{CC}^{air\,conditioner} \frac{q_{CU.CC}^{Raw}}{G_{CC}} + N_{CC}^{illuminations} n_{CC}^{illuminations} \frac{q_{CU.CC}^{Raw}}{G_{CC}} \tag{21}$$

Equation (22) provides the energy consumed at the disassembly center.

$$E^{DC} = N_{DC}^{air\ conditioner} n_{DC}^{air\ conditioner} \frac{q_{CC.DC}^{Raw}}{G_{DC}} + N_{DC}^{illuminations} n_{DC}^{illuminations} \frac{q_{CC.DC}^{Raw}}{G_{DC}} \qquad (22)$$

Equation (23) provides the energy consumed at the waste center.

$$E^{WC} = N_{WC}^{air\ conditioner} n_{WC}^{air\ conditioner} \frac{q_{DC.WC}^{Raw}}{G_{WC}} + N_{WC}^{illuminations} n_{WC}^{illuminations} \frac{q_{DC.WC}^{Raw}}{G_{WC}} \qquad (23)$$

Equation (24) provides the energy consumed at the refurbishing center.

$$E^{RC} = N_{RC}^{air\ conditioner} n_{RC}^{air\ conditioner} \frac{q_{DC.RC}^{Raw}}{G_{RC}} + N_{RC}^{illuminations} n_{RC}^{illuminations} \frac{q_{DC.RC}^{Raw}}{G_{RC}} \qquad (24)$$

**Total carbon emission $f_3$**

The third objective is to minimize carbon emission. The total emission is comprised of two components, that are, emission from plants and emission during transportation expressed in Eq. (25):

$$\text{Minimize} f_3 = em_{PL} + em^T \qquad (25)$$

The quantity of $CO_2$ which is emitted from the plant during production can be computed by using Eq. (26).

$$em_{PL} = \alpha_{PL_j} E_{PL_j}^{machine} q_{PL_j}^{Raw} + \alpha_{PL_j} E_{PL_j}^{air\ copmressor} + \alpha_{PL_j} E_{PL_j}^{air\ conditioner} + \alpha_{PL_j} E_{PL_j}^{illuminations} \qquad (26)$$

The quantity of $CO$ emitted from vehicles during the transportation of material from supplier to plant and shipment of the product to the retailer, customer, collection center, disassembly center waste center, and refurbishing center can be calculated by using Eq. (27).

$$\begin{aligned}
em^T &= em_{SU.PL}^{T.Raw} + em_{PL.RE}^{T.Product} + em_{PL.WC}^{T.Product} \\
&\quad + em_{RE.CU}^{T.Product} + em_{CU.CC}^{T.Raw} + em_{CC.DC}^{T.Raw} \\
&\quad + em_{CC.PL}^{T.Raw} + em_{DC.WC}^{T.Raw} + em_{DC.RC}^{T.Raw} + em_{RC.PL}^{T.Raw} \\
&= \alpha^T \frac{q_{SU.PL}^{Raw}}{V} D_{SU.PL} + \alpha^T \frac{q_{PL.RE}^{Product}}{V} D_{PL.RE} + \alpha^T \frac{q_{PL.WC}^{Product}}{V} D_{PL.WC} \\
&\quad + \alpha^T \frac{q_{RE.CU}^{Product}}{V} D_{RE.CU} + \alpha^T \frac{q_{CU.CC}^{Raw}}{V} D_{CU.CC} + \alpha^T \frac{q_{CC.DC}^{Raw}}{V} D_{CC.DC} \\
&\quad + \alpha^T \frac{q_{CC.PL}^{Raw}}{V} D_{CC.PL} + \alpha^T \frac{q_{DC.WC}^{Raw}}{V} D_{DC.WC} + \alpha^T \frac{q_{DC.RC}^{Raw}}{V} D_{DC.RC} \\
&\quad + \alpha^T \frac{q_{RC.PL}^{Raw}}{V} D_{RC.PL}
\end{aligned} \qquad (27)$$

## Constraints
### Supply constraints

Equation (28) ensures that the quantity of material shipped to the plant, retailer, customer, collection center and refurbishing center, or disassembly center cannot be greater than their capacity.

$$
\begin{aligned}
\sum_{PL} q_{SU.PL}^{Raw} &\leq Ca_{SU} \\
\sum_{RE} q_{PL.RE}^{Product} + \sum_{WC} q_{PL.WC}^{Product} &\leq Ca_{PL} \\
\sum_{CC} q_{RE.CC}^{Raw} &\leq Ca_{RE} \\
\sum_{PL} q_{CC.PL}^{Raw} + \sum_{DC} q_{PL.DC}^{Raw} &\leq Ca_{DC} \\
\sum_{RC} q_{DC.RC}^{Raw} + \sum_{WC} q_{DC.WC}^{Raw} &\leq Ca_{DC} \\
\sum_{PL} q_{RC.PL}^{Raw} &\leq Ca_{RC}
\end{aligned}
\tag{28}
$$

### Demand constraints

Equation (29) ensure that the demands of the plant, retailer, and customer are fulfilled, respectively.

$$
\begin{aligned}
\sum_{SU} q_{SU.PL}^{Raw} &\geq D_{PL} \\
\sum_{PL} q_{PL.RE}^{Product} &\geq D_{RE} \\
\sum_{RE} q_{RE.CU}^{Product} &\geq D_{CU}
\end{aligned}
\tag{29}
$$

### Balance equation

Equation (31) represent the balanced constrained equation.

$$
\begin{aligned}
\sum_{SU} q_{SU.PL}^{Raw} &= \sum_{RE} q_{PL.RE}^{Product} + \sum_{WC} q_{PL.WC}^{Product} \\
\sum_{PL} q_{PL.RE}^{Product} &= \sum_{CU} q_{RE.CU}^{Product} \\
\sum_{RE} q_{RE.CU}^{Product} &= \sum_{CC} q_{CU.CC}^{Product} \\
\sum_{CU} q_{CU.CC}^{Raw} &= \sum_{PL} q_{CC.PL}^{Raw} + \sum_{DC} q_{CC.DC}^{Raw} \\
\sum_{CC} q_{CC.DC}^{Raw} &= \sum_{WC} q_{DC.WC}^{Raw} + \sum_{RC} q_{DC.RC}^{Raw} \\
\sum_{DC} q_{DC.RC}^{Raw} &= \sum_{PL} q_{RC.PL}^{Raw}
\end{aligned}
\tag{30}
$$

***Informational constraints***

$$(1 - \beta_{PL_j}^{machine} \, q_{PL_j}^{Raw}) \geq q_{PL_{j+1}}^{Raw}$$

$$\gamma_{PL_j}^{air\ conditioner} \, n_{PL_j}^{air\ conditioner} \geq n_{PL_j}^{machine}$$

$$\gamma_{PL_j}^{air\ compressor} \, n_{PL_j}^{air\ compressor} \geq n_{PL_j}^{machine}$$

$$n_{PL_j}^{illuminations} \geq \gamma_{PL_j}^{illuminations} n_{PL_j}^{machine}$$

$$\Re_{PL_j}^{machine} n_{PL_j}^{machine} \geq q_{PL_{j+1}}^{Raw}$$

$$Ca_{RE} n_{RE}^{air\ conditioner} \geq \gamma_{RE}^{air\ conditioner} q_{PL.RE}^{Product}$$

$$Ca_{RE} n_{RE}^{illuminations} \geq \gamma_{RE}^{illuminations} q_{PL.RE}^{Product}$$

$$Ca_{CC} n_{CC}^{air\ conditioner} \geq \gamma_{CC}^{air\ conditioner} q_{CU.CC}^{Raw}$$

$$Ca_{CC} n_{CC}^{illuminations} \geq \gamma_{CC}^{illuminations} q_{CU.CC}^{Raw}$$

$$Ca_{DC} n_{DC}^{air\ conditioner} \geq \gamma_{DC}^{air\ conditioner} q_{CC.DC}^{Raw}$$

$$Ca_{DC} n_{DC}^{illuminations} \geq \gamma_{DC}^{illuminations} q_{CC.DC}^{Raw}$$

$$Ca_{RC} n_{RC}^{air\ conditioner} \geq \gamma_{RC}^{air\ conditioner} q_{DC.RC}^{Raw}$$

$$Ca_{RC} n_{RC}^{illuminations} \geq \gamma_{RC}^{illuminations} q_{DC.RC}^{Raw}$$

$$(31)$$

***Transshipment constraints***

$$\sum_{CU} q_{CU.CC}^{Raw} \geq R_1^{Rate} \sum_{RE} q_{RE.CU}^{Product}$$

$$\sum_{CU} q_{CU.CC}^{Raw} \geq R_2^{Rate} \sum_{CC} q_{CC.DC}^{Raw}$$

$$\sum_{CU} q_{CU.CC}^{Raw} \geq R_3^{Rate} \sum_{CC} q_{CC.PL}^{Raw}$$

$$\sum_{CC} q_{CC.DC}^{Raw} \geq R_4^{Rate} \sum_{DC} q_{DC.RC}^{Raw}$$

$$\sum_{CC} q_{CC.DC}^{Raw} \geq R_5^{Rate} \sum_{DC} q_{DC.WC}^{Raw}$$

$$(32)$$

## SOLUTION METHODOLOGY

The developed multi-objective linear programming model is solved by using a four-valued refined neutrosophic optimization technique (FVRNOT) proposed by *Freen et al. (2020)*. A four-valued refined neutrosophic set is one of several types of neutrosophic sets in which indeterminacy is divided into $\delta =$ uncertainty and $\eta =$ contradiction, where $\eta = \mu \wedge v$. The values of $\mu, \delta, \eta$ *and* $v$ fall into the range $[0, 1]$, and $0 \leq \mu + \delta + \eta + v \leq 4$. Mathematical expression of the set in given in Eq. (33).

$$Z^{\sim RN} = \{(s, \mu_{Z^{\sim RN}}(s), \delta_{Z^{\sim RN}}(s), \eta_{Z^{\sim RN}}(s), v_{Z^{\sim RN}}(s)) : s \in S\}. \qquad (33)$$

The objective functions are transformed into neutrosophic fuzzy constraints when optimization is executed using FVRNOT. The uncertainty is effectively handled by this

four-valued refined neutrosophic model that has been established and enhanced. The multi-objective four-valued refined neutrosophic linear programming model that is being described can handle uncertain data, which avoids unsustainable modeling. The computational algorithm of the stated approach includes the following steps:

1. One objective function from a set of $N$ objectives should be used to solve the first objective function. Under the stated constraints, the values of the decision variables and objective functions will be determined.

2. Now, through using decision variables from step 1, calculate the values of the undetermined objective, *i.e.*, $(N - 1)$.

3. Steps 1 and 2 must be repeated for the rest $(N - 1)$ objective functions to get the pay-off matrix 34.

$$
\begin{bmatrix}
f_1^*(s^1) & f_2(s^1) & \cdots & f_n(s^1) \\
f_1(s^2) & f_2^*(s^2) & \cdots & f_n(s^2) \\
\vdots & \vdots & \vdots & \vdots \\
f_1(s^n) & f_2(s^n) & \cdots & f_n^*(s^n)
\end{bmatrix}
\tag{34}
$$

4. For every objective function $f_p$ determine the lower bound $L_p$ and the upper bound $U_p$ such that

$$
L_p = \max_{r=1}^{n}\{f_p(s^r)\} \text{ and } U_p = \min_{r=1}^{n}\{f_p(s^r)\}.
$$

5. The lower and upper bounds for truth membership function $\mu$, falsity membership function $\nu$, uncertainty membership function $\delta$ and contradiction membership function $\eta$ of the objective function $f_p$ are computed in Eqs. (35)–(38).

$$
L_p^{\mu} = L_p \quad ; \quad U_p^{\mu} = U_p
\tag{35}
$$

$$
L_p^{\nu} = L_p^{\mu} + t_1(U_p^{\mu} - L_p^{\mu}) \quad ; \quad U_p^{\nu} = U_p^{\mu}
\tag{36}
$$

$$
L_p^{\delta} = L_p^{\mu} + t_2(U_p^{\mu} - L_p^{\mu}) \quad ; \quad U_p^{\delta} = U_p^{\mu}
\tag{37}
$$

$$
L_p^{\eta} = L_p^{\mu} \wedge L_p^{\nu} \quad ; \quad U_p^{\eta} = U_p^{\mu} \wedge U_p^{\nu} + t_3(U_p^{\mu} \wedge U_p^{\nu} - L_p^{\mu} \wedge L_p^{\nu})
\tag{38}
$$

where $t_1, t_2, t_3 \in (0, 1)$.

6. Define the truth, uncertain, false, and contradiction membership functions as in Eqs. (39)–(42)

$$
\mu_p(f_p(s)) = \begin{cases}
1 & \text{if } f_p(s) \leq L_p^{\mu}, \\
\frac{U_p^{\mu} - f_p(s)}{U_p^{\mu} - L_p^{\mu}} & \text{if } L_p^{\mu} \leq f_p(s) \leq U_p^{\mu}, \\
0 & \text{if } f_p(s) \geq U_p^{\mu},
\end{cases}
\tag{39}
$$

$$
\delta_p(f_p(s)) = \begin{cases}
1 & \text{if } f_p(s) \leq L_p^{\delta}, \\
\frac{U_p^{\delta} - f_p(s)}{U_p^{\delta} - L_p^{\delta}} & \text{if } L_p^{\delta} \leq f_p(s) \leq U_p^{\delta}, \\
0 & \text{if } f_p(s) \geq U_p^{\delta},
\end{cases}
\tag{40}
$$

$$v_p(f_p(s)) = \begin{cases} 0 & \text{if } f_p(s) \leq L_p^v, \\ \frac{f_p(s) - L_p^v}{U_p^v - L_p^v} & \text{if } L_p^v \leq f_p(s) \leq U_p^v, \\ 1 & \text{if } f_p(s) \geq U_p^v, \end{cases} \tag{41}$$

$$\eta_p(f_p(s)) = \begin{cases} 1 & \text{if } f_p(s) \leq L_p^\eta, \\ \frac{U_p^\eta - f_p(s)}{U_p^\eta - L_p^\eta} & \text{if } L_p^\eta \leq f_p(s) \leq U_p^\eta, \\ 0 & \text{if } f_p(s) \geq U_p^\eta, \end{cases} \tag{42}$$

7. Therefore, after finding the membership and non-membership functions, an equivalent crisp optimization problem is formed in Equation (43).

$$\max(\phi - \sigma + \psi + \theta), \tag{43}$$

such that

$$\mu_p(f_p(s)) \geq \phi, \quad v_p(f_p(s)) \leq \sigma, \quad \delta_p(f_p(s)) \geq \psi, \quad \eta_p(f_p(s)) \geq \theta \tag{44}$$

with

$$0 \leq \phi + \sigma + \psi + \theta \leq 4 \text{ and } \phi \geq \sigma, \psi, \theta \tag{45}$$

and subject to the original constraints.

## ASSESSMENT: A CASE STUDY

A real case study is employed to determine the effectiveness and validity of the previously mentioned designed fuzzy multi-objective optimization model. For this purpose data were collected from already conducted research studies and minor changes were made to make that data suitable for practice. The plant uses several distinct procedures to make its products, thus different phases require varied numbers of machines, air conditioners, air compressors, and illumination. Additionally, each center in the suggested model is made up of such equipment, and each piece of equipment has a varied mass, energy consumption, and $CO_2$ emission quantity.

The data for a manufacturing system that produces, stores, and transports the product is shown in Table 3. The supply chain in this instance is facilitated by two different renewable energy sources, particularly oil, which is used as direct sources to generate thermal energy and electricity, respectively; illustrating how to create a sustainable manufacturing system. The solution methodology defined in the previous section is carried out in the following manner.

Step 1–3: The three objective functions are simultaneously solved, and the values of decision variables are substituted into the objective functions that provide the pay-off matrix presented in Eq. (46).

$$\begin{bmatrix} 6200100 & 14806000 & 616040 \\ 6200100 & 13746 & 24350 \\ 6200100 & 13746 & 24350 \end{bmatrix} \tag{46}$$

**Table 3 Data table.**

| Amount of $CO_2$ emission factor per kWh using oil (per mile) | | | |
|---|---|---|---|
| $\alpha_{PL_j}$ | 0.04 kg/kWh | $\alpha^T$ | 0.41 kg/mile |
| **Cost** | | | |
| $C_{PL}^{Fixed}$ | 6,000,000 \$ | $C_{SU.PL}^{Raw}$ | 2 \$ |
| $C_{PL.RE}^{Product}$ | 3 \$ | $C_{PL.RE}^{I}$ | 2 \$ |
| $C_{CL}^{T}$ | 2 \$ | | |
| **Distance in miles** | | | |
| $D_{SU.PL}$ | 50 | $D_{PL.RE}$ | 30 |
| $D_{PL.WC}$ | 05 | $D_{RE.CU}$ | 15 |
| $D_{CU.CC}$ | 10 | $D_{CC.PL}$ | 15 |
| $D_{CC.DC}$ | 20 | $D_{DC.WC}$ | 08 |
| $D_{DC.RC}$ | 17 | $D_{RC.PL}$ | 20 |
| **Installed power in kW** | | | |
| $N_{PL_j}^{machine}$ | 180 | $N_{PL_j}^{air\ compressor}$ | 150 |
| $N_{PL_j}^{air\ conditioner}$ | 2 | $N_{PL_j}^{illuminations}$ | 0.5 |
| $N_{RE}^{air\ conditioner}$ | 0.5 | $N_{RE}^{illuminations}$ | 0.1 |
| $N_{CC}^{air\ conditioner}$ | 2 | $N_{CC}^{illuminations}$ | 0.3 |
| $N_{DC}^{air\ conditioner}$ | 1.5 | $N_{DC}^{illuminations}$ | 0.2 |
| $N_{RC}^{air\ conditioner}$ | 1 | $N_{RC}^{illuminations}$ | 0.1 |
| **Capacity** | | | |
| V (per vehicle) | 20,000 kg | $\rho_{PL_j}^{air\ compressor}$ | 666 $m^3/h$ |
| $Ca_{PL}$ | 16,000 kg/month | $Ca_{SU}$ | 30,000 kg/month |
| $Ca_{RE}$ | 14,000 kg/month | $Ca_{CC}$ | 5,000 kg/month |
| $Ca_{DC}$ | 3,000 kg/month | $Ca_{RC}$ | 2,000 kg/month |
| **Covering rate (units) used in each center** | | | |
| $\gamma_{PL_j}^{air\ conditioner}$ | 1.5 | $\gamma_{PL_j}^{air\ compressor}$ | 03 |
| $\gamma_{PL_j}^{illuminations}$ | 15 | $\gamma_{RE}^{air\ conditioner}$ | 1.5 |
| $\gamma_{RE}^{illuminations}$ | 10 | $\gamma_{CC}^{air\ conditioner}$ | 1.5 |
| $\gamma_{CC}^{illuminations}$ | 10 | $\gamma_{DC}^{air\ conditioner}$ | 1.6 |
| $\gamma_{DC}^{illuminations}$ | 9 | $\gamma_{RC}^{air\ conditioner}$ | 1.4 |
| $\gamma_{RC}^{illuminations}$ | 10 | | |
| **Parameters involved in manufacturing** | | | |
| $\beta_{PL_j}^{machine}$ | 0.001 | $\Re_{PL_j}$ | 1840 |
| $\mu_{PL_j}$ | 0.6 | $R_1^{Rate}$ | 0.07 |
| $R_2^{Rate}$ | 0.02 | $R_3^{Rate}$ | 0.05 |
| $R_4^{Rate}$ | 0.10 | $R_5^{Rate}$ | 0.20 |
| **Quantity of material/product produced and the demand for each center (kg/month)** | | | |
| $G_{PL}$ | 751,569 | $G_{RE}$ | 632,650 |
| $G_{CC}$ | 53,164 | $G_{DC}$ | 514,500 |

(Continued)

| Table 3 (continued) | | | |
|---|---|---|---|
| $G_{WC}$ | 50,198 | $G_{RC}$ | 495,045 |
| $D_{PL}$ | 15,000 | $D_{RE}$ | 12,000 |

Step 4–5: The lower and upper bounds for truth membership function $\mu$, falsity membership function $v$, uncertainty membership function $\delta$ and contradiction membership function $\eta$ of the objective function $f_p$ are computed by using Eqs. (35)–(38).

For total cost investment $f_1$

$U_1^v = U_1^\mu = 6200100,$
$L_1^v = 6200100 + 0.3 \cdot 0 = 6200100,$
$U_1^\delta = U_1^\mu = 6200100, \quad L_1^\delta = 6200100 + 0.3 \cdot 0 = 6200100,$
$L_1^\eta = 6200100 \wedge 6200100 = 6200100, \quad U_1^\eta = 6200100.$

For total energy consumption $f_2$

$L_2^\mu = 13746, \quad U_2^\mu = 14806000,$
$U_2^v = U_2^\mu = 14806000, \quad L_2^v = 4451400,$
$U_2^\delta = 5930600, \quad L_2^\delta = 13746,$
$L_2^\eta = 14806000, \quad U_2^\eta = 8889000.$

For total carbon emission $f_3$

$L_3^\mu = 24350, \quad U_3^\mu = 616040,$
$U_3^v = 616040, \quad L_3^v = 201900,$
$U_3^\delta = 261020, \quad L_3^\delta = 24350,$
$L_3^\eta = 616040, \quad U_3^\eta = 379360.$

Step 6: Truth, uncertain, falsity, and contradiction functions are defined by using the above bounds and Eqs. (39)–(42).

Step 7: Single objective optimization problem is defined as in Eq. (43) subject to constraints in Eqs. (28)–(32), (44) and (45).

## RESULTS

The specified multi-objective optimization model for a sustainable manufacturing system was solved and subsequently optimized using FVRNOT. Solutions appear to have a maximum range for the number of machines $n_{PL_j}^{machine} = 7$, number of air conditioner units $n_{PL_j}^{air\ conditioner} = 5$, number of illuminations $n_{PL_j}^{illuminations} = 105$ and the number of air compressor $n_{PL_j}^{air\ conditioner} = 3$ that are involved in the plant for manufacturing products; where j = 1 is fixed. In the given system for the retailer center, the number of air conditioner units is found $n_{RE}^{air\ conditioner} = 196$ and the number of illumination is

$n_{RE}^{illuminations} = 1,305$. In the collection center $n_{CC}^{air\ conditioner} = 13$ and $n_{CC}^{illuminations} = 86$. The result also depicts the ideal supply chain strategy for the raw material, that is, $q_{SU.PL}^{Raw} = 12,000\ kg/month$ and the quantity of manufactured product transported from the plant to the retailer center $q_{PL.RE}^{Product} = 12,000\ kg/month$. This amount fulfilled the consumer demands, which is $q_{RE.CU}^{Product} = 12,000\ kg/month$. The amount of material that the customer has supplied to the collection center is evaluated as $q_{CU.CC}^{Raw} = 840\ kg/month$. By using the lower and upper bound of each function the aspiration level is determined. Considering the objective function of $CO_2$ emission whose optimal value is 30,646 kg, the aspiration level can be calculated as:

$$\text{Aspiration level} = \frac{U_3^\mu - f_3(s)}{U_3^\mu - L_3^\mu} \times 100 = \frac{616040 - 30646}{616040 - 24350} \times 100 = 98.9\% \tag{47}$$

Comparably, we will determine the aspiration level of the first two functions. The ultimate satisfaction level is attained in the minimization scenario when the resulting value approaches the lower bounds. When the optimized value is identical to the lower bounds, the level of satisfaction is 100%.

The result indicates that the number of machines and equipment should be carefully chosen to maximize production efficiency while minimizing energy consumption and operational costs. These quantities are indicative of a system that balances production capacity with sustainability goals, avoiding overuse of resources and potential environmental impact. The high number of air conditioners and illuminations suggests a focus on maintaining product quality and visibility. However, this also implies substantial energy consumption. The design should consider energy-efficient technologies and practices to reduce the environmental footprint while meeting operational needs. This configuration supports effective raw material handling while managing energy use. The focus on minimizing energy consumption in the collection center aligns with sustainability objectives and reduces overall operational costs. The efficient flow of materials and products ensures that consumer demand is met without excess or shortage, contributing to a balanced and responsive supply chain. This strategy minimizes waste and enhances the overall efficiency of the closed-loop system. The high aspiration level reflects the system's effectiveness in minimizing CO emissions, which is crucial for sustainability. This result underscores the importance of integrating emission reduction strategies into the design and operation of manufacturing systems.

## Comparison

The under-consideration optimization problem is solved by using the intuitionistic fuzzy optimization technique (IFOT) and neutrosophic optimization technique (NOT) method.

An intuitionistic fuzzy set is defined as $Z^{\sim IFS} = \{(s, \mu_{Z^{\sim IFS}}(s), v_{Z^{\sim IFS}}(s)) : s \in S\}$. The intuitionistic fuzzy optimization technique (IFOT) is based on pay-off matrix, Eqs. (35), (36), (39) and (41). Where $\max(\phi - \sigma)$ is fuzzy objective and $\mu_p(f_p(s)) \geq \phi$, $v_p(f_p(s)) \leq \sigma$, $0 \leq \phi + \sigma \leq 1$ and $\phi \geq \sigma$ are fuzzy constraints for IFOT.

**Table 4 Comparison of techniques.**

| Techniques | Cost | Energy consumption | $CO_2$ emission |
|---|---|---|---|
| IFOT | 6,200,100 $ | 2,554,000 kWh | 125,960 kg |
| NOT | 6,200,100 $ | 2,554,500 kWh | 125,980 kg |
| FVRNOT | 6,200,100 $ | 171,140 kWh | 30,646 kg |

Similarly, a neutrosophic set is defined as $Z^{\sim NS} = \{(s, \mu_{Z^{\sim NS}}(s), \delta_{Z^{\sim NS}}(s), v_{Z^{\sim NS}}(s)) : s \in S\}$. The neutrosophic optimization technique (NOT) is based on pay-off matrix, Eqs. (35)–(37), (39), (40) and (41). Where $\max(\phi - \sigma + \psi)$ is neutrospohic objective function and $\mu_p(f_p(s)) \geq \phi, \quad v_p(f_p(s)) \leq \sigma, \quad \delta_p(f_p(s)) \geq \psi, 0 \leq \phi + \sigma + \psi \leq 3$ and $\phi \geq \sigma, \psi$ are neutrosophic constraints. The values of the objective functions with the three techniques are presented in Table 4. It is clear that the FVRNOT method produces the best overall results compared to the other two methods because it requires less overall investment in terms of costs, energy use by the machines, and overall $CO_2$ emissions from all sources.

## Sensitivity analysis

The parameters such as inventory, production, and capacity of the multi-objective optimization model can directly be influenced and controlled by either operations or overall supply chain management. In addition, parameters like market demand and fixed return rate are mostly governed by market factors with significant impacts on the solution approach in the model. In this research, the parameters dependent on the market are selected to perform sensitivity analysis to have an insight into their influence on the objective functions. A custom experimental design is constructed based on the chosen factors and levels to analyze, and to run the model for evaluation of all scenarios, and the respective responses are illustrated in Table 5. The sensitivity analysis is performed by keeping one parameter constant and assessing the impact of one factor at all levels. For each level from 10% to +10%, this approach is repeated systematically. Further experiments repeatedly revealed similar trends of increasing or decreasing function values, resulting in five levels being added for explanatory purposes.

In Figs. 4 and 5, a 5% increase and 10% decrease in demand respectively, with the percentage change in demand used as a changing parameter the objective function's values are calculated at all levels. Due to increased demand, the additional use of machinery, transportation, and refurbishing is increasing energy consumption and $CO_2$ emission with a slight change in cost. Reducing demand leads to decreases in both energy consumption and $CO_2$ emissions. However, these changes in demand do not substantially affect cost, which remains nearly constant, highlighting a potential challenge for cost-sensitive strategies that aim to influence demand fluctuations. Similarly, the same trend at levels such as 10%, −5%, and +10% is clearly shown in Table 5.

Figures 6 and 7 demonstrate that 10% an increase and 10% decrease in return rate respectively, with changes in the return rate from −10% to +10% having a minimal impact

**Table 5 Experimental data for sensitivity analysis.**

| Levels | Demand | Return rate | Total cost | Energy consumption | $CO_2$ Emission |
|---|---|---|---|---|---|
| −10% | −10% | −10% | 6,180,090 | 1.55E+05 | 2.76E+04 |
| | −10% | −5% | 6,180,090 | 1.55E+05 | 2.76E+04 |
| | −10% | 0% | 6,180,090 | 1.55E+05 | 2.76E+04 |
| | −10% | 5% | 6,180,090 | 1.55E+05 | 2.76E+04 |
| | −10% | 10% | 6,180,090 | 1.55E+05 | 2.76E+04 |
| | −10% | −10% | 6,180,090 | 1.55E+05 | 2.76E+04 |
| | −5% | −10% | 6,190,095 | 1.63E+05 | 2.91E+04 |
| | 0% | −10% | 6,200,100 | 1.71E+05 | 3.06E+04 |
| | 5% | −10% | 6,210,105 | 1.79E+05 | 3.22E+04 |
| | 10% | −10% | 6.22E+06 | 1.89E+05 | 3.37E+04 |
| −5% | −5% | −10% | 6,190,095 | 1.63E+05 | 2.91E+04 |
| | −5% | −5% | 6,190,095 | 1.63E+05 | 2.91E+04 |
| | −5% | 0% | 6,190,095 | 1.63E+05 | 2.91E+04 |
| | −5% | 5% | 6,190,095 | 1.63E+05 | 2.91E+04 |
| | −5% | 10% | 6,190,095 | 1.63E+05 | 2.91E+04 |
| | −10% | −5% | 6,180,090 | 1.55E+05 | 2.76E+04 |
| | −5% | −5% | 6,190,095 | 1.63E+05 | 2.91E+04 |
| | 0% | −5% | 6,200,100 | 1.71E+05 | 3.06E+04 |
| | 5% | −5% | 6,210,105 | 1.79E+05 | 3.22E+04 |
| | 10% | −5% | 6,220,110 | 1.89E+05 | 3.37E+04 |
| 0% | 0% | −10% | 6,200,100 | 1.71E+05 | 3.06E+04 |
| | 0% | −5% | 6,200,100 | 1.71E+05 | 3.06E+04 |
| | 0% | 0% | 6,220,110 | 1.71E+05 | 3.06E+04 |
| | 0% | 5% | 6,200,100 | 1.71E+05 | 3.06E+04 |
| | 0% | 10% | 6200,100 | 1.71E+05 | 3.06E+04 |
| | −10% | 0% | 6,180,090 | 1.55E+05 | 2.76E+04 |
| | −5% | 0% | 6,190,095 | 1.63E+05 | 2.91E+04 |
| | 0% | 0% | 6,200,100 | 1.71E+05 | 3.06E+04 |
| | 5% | 0% | 6,210,105 | 1.79E+05 | 3.22E+04 |
| | 10% | 0% | 6,220,110 | 1.89E+05 | 3.37E+04 |
| 5% | 5% | −10% | 6,210,105 | 1.79E+05 | 3.22E+04 |
| | 5% | −5% | 6,210,105 | 1.79E+05 | 3.22E+04 |
| | 5% | 0% | 6,210,105 | 1.79E+05 | 3.22E+04 |
| | 5% | 5% | 6,210,105 | 1.79E+05 | 3.22E+04 |
| | 5% | 10% | 6,210,105 | 1.79E+05 | 3.22E+04 |
| | −10% | 5% | 6,180,090 | 1.55E+05 | 2.76E+04 |
| | −5% | 5% | 6,190,095 | 1.63E+05 | 2.91E+04 |
| | 0% | 5% | 6,200,100 | 1.71E+05 | 3.06E+04 |
| | 5% | 5% | 6,210,105 | 1.79E+05 | 3.22E+04 |
| | 10% | 5% | 6,220,110 | 1.89E+05 | 3.37E+04 |

(Continued)

| Table 5 (continued) | | | | | |
|---|---|---|---|---|---|
| Levels | Demand | Return rate | Total cost | Energy consumption | $CO_2$ Emission |
| 10% | 10% | −10% | 6,220,110 | 1.89E+05 | 3.37E+04 |
| | 10% | −5% | 6,220,110 | 1.89E+05 | 3.37E+04 |
| | 10% | 0% | 6,220,110 | 1.89E+05 | 3.37E+04 |
| | 10% | 5% | 6,220,110 | 1.89E+05 | 3.37E+04 |
| | 10% | 10% | 6,220,110 | 1.89E+05 | 3.37E+04 |
| | −10% | 10% | 6,180,090 | 1.55E+05 | 2.76E+04 |
| | −5% | 10% | 6,190,095 | 1.63E+05 | 2.91E+04 |
| | 0% | 10% | 6,200,100 | 1.71E+05 | 3.06E+04 |
| | 5% | 10% | 6,210,105 | 1.79E+05 | 3.22E+04 |
| | 10% | 10% | 6,220,110 | 1.89E+05 | 3.37E+04 |

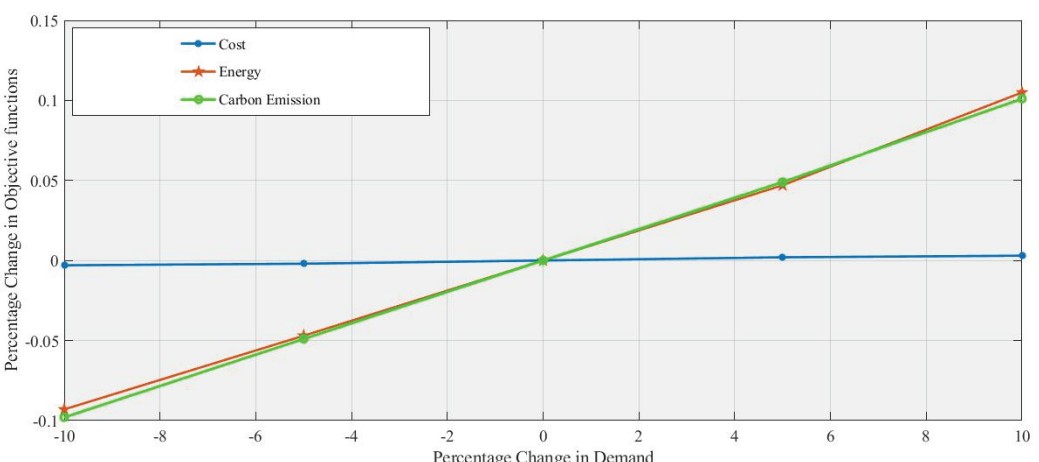

**Figure 4 Effect of 5% increase in demand on objective functions values.**

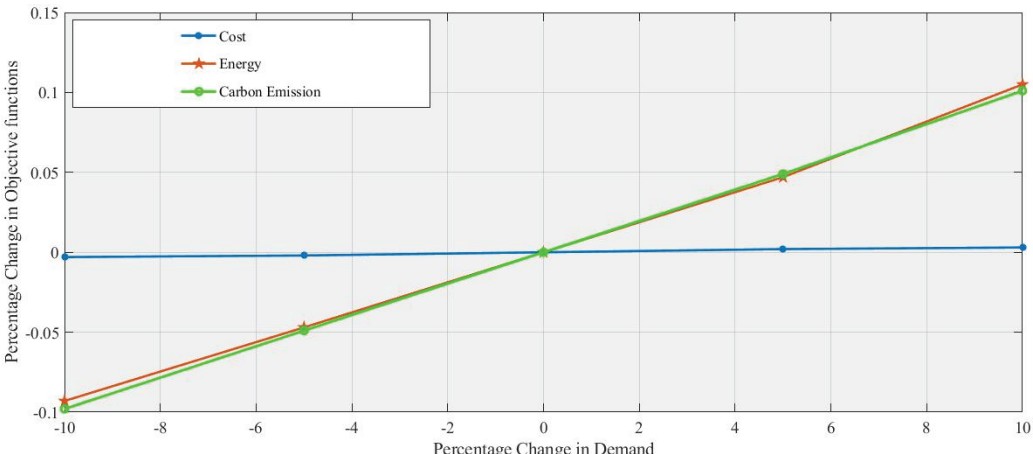

**Figure 5 Effect of 10% decrease in demand on objective functions values.**

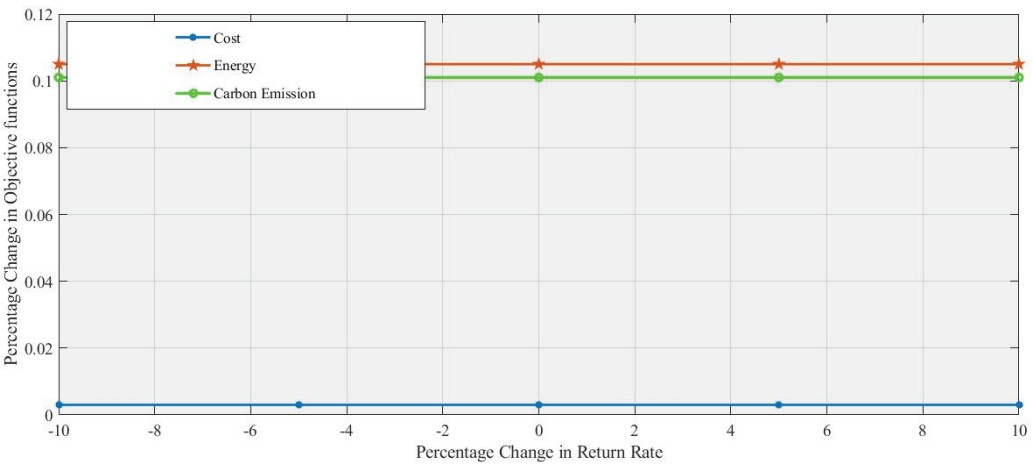

**Figure 6  Effect of 10% increase in return rate on objective functions values.**

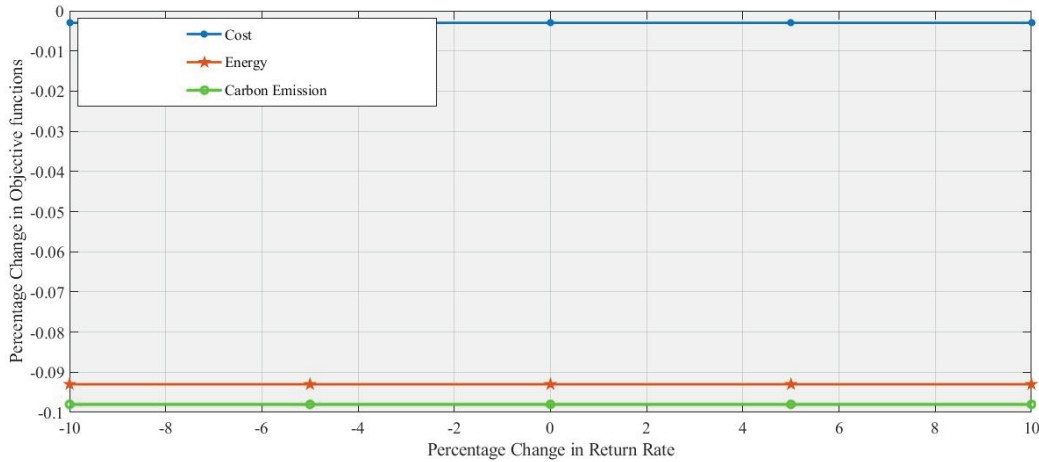

**Figure 7  Effect of 10% decrease in return rate on objective functions values.**

on the performance of all objective functions. These objective functions exhibit a similar, stable response in the range of changes in the return rate, which indicates that these factors (cost, energy, and $CO_2$ emission) are relatively insensitive to changes in return. Energy and Carbon Emissions show slight deviations, with minor decreases at lower return rates and small increases at higher rates, though the changes are minimal.

## MANAGERIAL INSIGHTS

Professionals unfamiliar with the complicated mathematical framework need to concentrate on understanding the primary decisions optimized in the CLMS design. The model considers economic and environmental factors when optimizing production, distribution, inventory, recycling, and location decisions. By minimizing the system's total cost, and energy consumption, and lowering $CO_2$ emissions, the model seeks to advance both environmental and economic sustainability within the CLMS.

In this article, manufacturing, transportation, recycling, and location problems for CLMS are introduced. The managers of supply chains can plan the placement of their facilities to maximize consumer demand with the aid of the model. The methodology minimizes overall supply chain costs, and decisions about production, distribution, inventory holding, recycling, and location will increase profits for those involved in the supply chain in achieving financial sustainability for their companies. Supply chain managers can also use this model to minimize the quantity of $CO_2$ emissions emitted during transportation and to guarantee that products are moved properly across the CLMS. As a result, an industrial system can use the planned sustainable CLMS network.

The fuzzy multiobjective optimization model provides valuable results for managers to increase the stability of production. It assists managers in making better decisions despite uncertainties in parameters such as demand and regulations. This research shows how closed-loop systems need to be adopted in industries because of their benefits in reducing waste and cost savings due to recycling and reusable materials implementation. Management is urged to adopt the principles of circular economy to better resource utilization and supply chain resilience. Meanwhile, this study also demonstrates that the impact of sustainability on manufacturing processes changes competitive advantages in the long perspective, enhances customer loyalty, and helps to meet environmental requirements. For solving the issues of sustainable development, resource use, and waste reduction along the principles of the circular economy, the study presents practitioners and managers with operational tools and tactics to make their activities sustainable and cost-effective even in challenging environments.

## CONCLUSION

In designing supply chains, manufacturers increasingly prioritize analytics that balance system efficiency with sustainability. This research contributes to this field by exploring sustainable development in closed-loop production processes and product transportation across all stages of the supply chain. It introduces a refined fuzzy multi-objective model for optimizing a green production framework, carefully considering various constraints such as cost, energy consumption, and emissions. By addressing these competing objectives, the proposed model offers an inclusive tool for manufacturers to achieve sustainable production while minimizing overall investment costs, energy usage, and carbon emissions. The model was tested using data from a real-world scenario, successfully resolving the inherent uncertainties in demand and returns within a closed-loop manufacturing system. The model achieved a notable aspiration level of 98.9%, demonstrating its effectiveness in navigating the complexities of closed-loop supply chains. These findings provide supply chain designers with actionable insights and a benchmark for evaluating and implementing sustainability measures across various industries. Additionally, this research lays the foundation for optimizing production levels, determining optimal market pricing, and managing resource allocation in a circular economy framework.

However, this model has certain limitations. Firstly, its complexity, involving advanced mathematical terminologies and fuzzy logic, may limit its application to experts in the

field. Industries looking to adopt this model may need to hire skilled personnel or invest in training their existing staff, which can be time-consuming and costly. Moreover, while the model's theoretical robustness is evident, the study relied on hypothetical data based on previous research due to the unavailability of real-time data from existing industries. This reliance on secondary data highlights a gap in the direct application of the model to real-world industrial settings.

To further enhance the model's applicability and relevance, future research should focus on determining innovative strategies that incorporate environmental and social components into the optimization process more effectively. Practical evaluations of this model in real-world settings would provide valuable insights into its strengths and potential areas for refinement. Furthermore, expanding the scope of the study by integrating more objectives, such as waste management, customer satisfaction, and supply chain resilience, could yield a more complete framework for closed-loop supply chain management. Another critical area for future research is to explore the obstacles industries may face when implementing this model. For example, identifying barriers related to data collection and processing and the need for cross-sectoral collaboration can provide a clearer road map for operationalizing such models in practice. Overcoming these challenges will validate the model's effectiveness and ensure that it can be adapted to different industrial contexts, making a significant contribution to the advancement of sustainable supply chain management.

### Funding

This work was supported by the National Research Foundation of Korea (NRF) grant funded by the Korea government (MSIT) (No. RS-2023-00277907) and by the Technology Development Program of MSS (No. S3033853). The funders had no role in study design, data collection and analysis, decision to publish, or preparation of the manuscript.

### Grant Disclosures

The following grant information was disclosed by the authors:
National Research Foundation of Korea (NRF) grant funded by the Korea government (MSIT): RS-2023-00277907.
Technology Development Program of MSS: S3033853.

### Competing Interests

The authors declare that they have no competing interests.

### Author Contributions

- Sajida Kousar conceived and designed the experiments, analyzed the data, prepared figures and/or tables, and approved the final draft.
- Asma Alvi conceived and designed the experiments, analyzed the data, prepared figures and/or tables, authored or reviewed drafts of the article, and approved the final draft.

- Nasreen Kausar conceived and designed the experiments, authored or reviewed drafts of the article, and approved the final draft.
- Harish Garg performed the experiments, authored or reviewed drafts of the article, and approved the final draft.
- Seifedine Kadry performed the computation work, authored or reviewed drafts of the article, and approved the final draft.
- Jungeun Kim performed the experiments, authored or reviewed drafts of the article, and approved the final draft.

## Data Availability

The data is available in Table 3. The codes are available in the Supplemental Files.

## Supplemental Information

Supplemental information for this article can be found online at http://dx.doi.org/10.7717/peerj-cs.2591#supplemental-information.

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
