# Peer review of "Fuzzy multi-objective optimization model to design a sustainable closed-loop manufacturing system"

_PeerJ Computer Science, doi:10.7717/peerj-cs.2591_

## Round 0.1 · original submission · Major Revisions

Dear authors,

Thank you for submitting your manuscript. Feedback from the reviewers is now available. It is not recommended that your article be published in its current format. However, we strongly recommend that you address the issues raised by the reviewers, especially those related to readability, experimental design and validity, and resubmit your paper after making the necessary changes. Before submitting the manuscript, following should also be addressed:

1- The research gaps and contributions should be clearly summarized in the introduction section. Please evaluate how your study is different from others.
2- Section numbering should be corrected.
3- The values for the parameters of the algorithms selected for comparison should be given.
4- The paper lacks the running environment, including software and hardware. The analysis and configurations of experiments should be presented in detail for reproducibility.
5- Advantages and disadvantages of the proposed and adapted method should be clarified. What are the limitations of the method(s) used in this paper?
6- Equations should be used with correct equation number. Please do not use “as follows”, “given as”, etc. Explanation of the equations should also be checked. All variables should be written in italic as in the equations. Their definitions and boundaries should be defined. Necessary references should be provided.

Best wishes,

Reviewer 1 ·

Basic reporting

This domain is written satisfactorily.

Experimental design

It has some novelty.

Validity of the findings

New findings are found.

Additional comments

Accept as it is.

Reviewer 2 ·

Basic reporting

The authors clearly write the article, but in the introduction section, they state, "There is still no definitive definition for sustainability in manufacturing firms," without providing any references. Additionally, they use the term "Republicans and Democrats," without clarifying its meaning.

Experimental design

The authors state they use a fuzzy multi-objective linear programming model that is solved by using a four-valued refined neutrosophic optimization technique (FVRNOT), but they don't describe clearly, especially in the case study, how to implement this method.

Validity of the findings

In the comparison section, the authors compare FVRNOT method with the IFO and the NOT method, but there is no explanation why they need to compare and how to do calculations for these two other methods.

Reviewer 3 ·

Basic reporting

The paper is well-written and easy to follow. The introduction provides sufficient background on the topic and clearly states the research questions. The literature review is comprehensive and relevant. The methodology is described in detail, and the results are presented clearly. The conclusion summarizes the main findings and their implications.

Experimental design

The research questions are well-defined, relevant, and meaningful. The study fills an identified knowledge gap in the literature on sustainable closed-loop manufacturing systems. The methods are appropriate for the research questions, and the data analysis is rigorous.

Validity of the findings

The findings are valid and reliable. The data are robust and the conclusions are well-supported by the results.

Additional comments

I have a few suggestions to improve the manuscript:

In the introduction, the authors should provide more context on the environmental impact of the manufacturing industry. For example, they could discuss the amount of waste generated by the industry, the amount of energy consumed, and the amount of greenhouse gas emissions produced.
In the literature review, the authors should discuss the limitations of previous studies on sustainable closed-loop manufacturing systems. This would help to highlight the contribution of the current study.
In the methodology section, the authors should provide more information on the data collection process. For example, they could discuss how they collected the data on the costs, energy consumption, and emissions of the different manufacturing processes.
In the results section, the authors should provide more interpretation of the findings. For example, they could discuss the implications of the findings for the design of sustainable closed-loop manufacturing systems.
In the conclusion, the authors should discuss the limitations of the study and suggest directions for future research.
I believe that these suggestions would improve the manuscript and make it even more valuable to the literature.

Reviewer 4 ·

Basic reporting

Please see the attached file

Experimental design

Please see the attached file

Validity of the findings

Please see the attached file

Additional comments

Please see the attached file

Annotated reviews are not available for download in order to protect the identity of reviewers who chose to remain anonymous.

---

## Round 0.2 · Minor Revisions

Dear authors,

Thank you for the revision. Your paper seems to be sufficiently improved. However, please make the necessary changes and additions suggested by Reviewer 4 and resubmit for final decision.

Best wishes,

Reviewer 3 ·

Basic reporting

Clear and unambiguous, professional English used throughout.

Experimental design

Research question well defined, relevant & meaningful. It is stated how research fills an identified knowledge gap

Validity of the findings

good

Reviewer 4 ·

Basic reporting

-

Experimental design

-

Validity of the findings

-

Additional comments

The authors have clarified my previous comments. Additional comment is related to Table 2: Suggest that the last row is to show how the authors' study differs from existing studies. Suggest to accept after the above revision.

---

## Round 0.3 · Minor Revisions

Dear Authors,

One of the reviewers did not respond to the last invitation. In addition, another reviewer has indicated that your paper requires minor revisions. We encourage you to address these concerns and criticisms of Reviewer 4 and to submit an updated version of your paper for review.

Best wishes,

Reviewer 4 ·

Basic reporting

see attached file

Experimental design

see attached file

Validity of the findings

see attached file

Additional comments

see attached file

Annotated reviews are not available for download in order to protect the identity of reviewers who chose to remain anonymous.

---

## Round 0.4 · accepted · Accept

Dear Authors,

I am grateful for your revised paper, which now addresses all of the reviewers' comments. I am pleased to inform you that the paper has been accepted for publication.

Best wishes,

Reviewer 4 ·

Basic reporting

N/A

Experimental design

N/A

Validity of the findings

N/A

Additional comments

The authors have clarified all my previous comments and thus it is my suggestion for acceptance.